# Ambient particulate matter pollution and adult hospital admissions for pneumonia in urban China: A national time series analysis for 2014 through 2017

Yaohua Tian[1,2,3], Hui Liu[3,4], Yiqun Wu[3], Yaqin Si[5], Man Li[3], Yao Wu[3], Xiaowen Wang[3], Mengying Wang[3], Libo Chen[5], Chen Wei[5], Tao Wu[3], Pei Gao[3,6]*, Yonghua Hu[3,4]*

1 Department of Maternal and Child Health, School of Public Health, Tongji Medical College, Huazhong University of Science and Technology, Wuhan, China, 2 Ministry of Education Key Laboratory of Environment and Health, and State Key Laboratory of Environmental Health (Incubating), School of Public Health, Tongji Medical College, Huazhong University of Science and Technology, Wuhan, China, 3 Department of Epidemiology and Biostatistics, School of Public Health, Peking University, Beijing, China, 4 Medical Informatics Center, Peking University, Beijing, China, 5 Beijing HealthCom Data Technology Co. Ltd, Beijing, China, 6 Key Laboratory of Molecular Cardiovascular (Peking University), Ministry of Education, Beijing, China

☯ These authors contributed equally to this work.
* yhhu@bjmu.edu.cn (YH); peigao@bjmu.edu.cn (PG)

**Data Availability Statement:** Air pollution data used in this study can be obtained from the China Environmental Monitoring Center (http://106.37. 208.233:20035). Meteorological data can be

## Abstract

### Background

The effects of ambient particulate matter (PM) pollution on pneumonia in adults are inconclusive, and few scientific data on a national scale have been generated in low- or middle-income countries, despite their much higher PM concentrations. We aimed to examine the association between PM levels and hospital admissions for pneumonia in Chinese adults.

### Methods and findings

A nationwide time series study was conducted in China between 2014 and 2017. Information on daily hospital admissions for pneumonia for 2014–2017 was collected from the database of Urban Employee Basic Medical Insurance (UEBMI), which covers 282.93 million adults. Associations of PM concentrations and hospital admissions for pneumonia were estimated for each city using a quasi-Poisson regression model controlling for time trend, temperature, relative humidity, day of the week, and public holiday and then pooled by random-effects meta-analysis. Meta-regression models were used to investigate potential effect modifiers, including cities' annual-average air pollutants concentrations, temperature, relative humidity, gross domestic product (GDP) per capita, and coverage rates by the UEBMI. More than 4.2 million pneumonia admissions were identified in 184 Chinese cities during the study period. Short-term elevations in PM concentrations were associated with increased pneumonia admissions. At the national level, a 10-µg/m³ increase in 3-day moving average (lag 0–2) concentrations of PM₂.₅ (PM ≤2.5 µm in aerodynamic diameter) and PM₁₀ (PM

accessed from the China Meteorological Data Sharing Service System (http://data.cma.cn/). Summarized health data can be accessed by contacting the National Insurance Claims for Epidemiological Research (NICER) Group, School of Public Health, Peking University; contact email, 0016156078@bjmu.edu.cn.

**Funding:** YH was supported by the National Natural Science Foundation of China (Grant No. 81872695), and PG was supported by the National Natural Science Foundation of China (Grant No. 91546120) and the National Thousand Talents Program for Distinguished Young Scholars, China (QNQR201501). The funders had no role in study design, data collection and analysis, decision to publish, or preparation of the manuscript.

**Competing interests:** The authors have declared that no competing interests exist.

**Abbreviations:** APHEA, Air Pollution and Health—A European Approach; CI, confidence interval; CO, carbon monoxide; df, degree of freedom; GDP, gross domestic product; ICD, International Classification of Diseases; $NO_2$, nitrogen dioxide; $O_3$, ozone; $PM_{2.5}$, particulate matter $\leq$2.5 μm in aerodynamic diameter; $PM_{10}$, particulate matter $\leq$10 μm in aerodynamic diameter; $SO_2$, sulfur dioxide; UEBMI, Urban Employee Basic Medical Insurance.

$\leq$10 μm in aerodynamic diameter) was associated with 0.31% (95% confidence interval [CI] 0.15%–0.46%, $P < 0.001$) and 0.19% (0.11%–0.30%, $P < 0.001$) increases in hospital admissions for pneumonia, respectively. The effects of $PM_{10}$ were stronger in cities with higher temperatures (percentage increase, 0.031%; 95% CI 0.003%–0.058%; $P = 0.026$) and relative humidity (percentage increase, 0.011%; 95% CI 0%–0.022%; $P = 0.045$), as well as in the elderly (percentage increase, 0.10% [95% CI 0.02%–0.19%] for people aged 18–64 years versus 0.32% [95% CI 0.22%–0.39%] for people aged $\geq$75 years; $P < 0.001$). The main limitation of the present study was the unavailability of data on individual exposure to PM pollution.

## Conclusions

Our findings suggest that there are significant short-term associations between ambient PM levels and increased hospital admissions for pneumonia in Chinese adults. These findings support the rationale that further limiting PM concentrations in China may be an effective strategy to reduce pneumonia-related hospital admissions.

## Author summary

### Why was this study done?

- Epidemiological studies have reported associations between short-term exposure to ambient particulate matter (PM) pollution and the risk of pneumonia.

- Previous studies have been primarily conducted in high-income countries, and the findings remain inconclusive.

- Few scientific data on a national scale have been generated in low- or middle-income countries, despite their much higher PM concentrations.

### What did the researchers do and find?

- We conducted a nationwide time series analysis using data on more than 4.2 million hospital admissions for pneumonia in 184 cities in China between 2014 and 2017 to estimate city-specific, national, and regional average associations between ambient PM pollution and pneumonia hospitalizations.

- Our results suggested that short-term increases in $PM_{2.5}$ and $PM_{10}$ were associated with increased hospital admissions for pneumonia. The effects of $PM_{10}$ were stronger in cities with higher temperatures and relative humidity, as well as in the elderly.

### What do these findings mean?

- To our knowledge, this is the first study in China to investigate the short-term associations of PM levels with hospital admissions for pneumonia on a national scale.

- Our findings support the rationale for further limiting PM concentrations in low- and middle-income countries.

## Introduction

Pneumonia is a major cause of mortality and morbidity worldwide [1, 2]. In China, an estimated 2.5 million pneumonia cases occur annually, and 5% of these individuals die of pneumonia-related illness [3]. Furthermore, pneumonia is closely related to complications such as pleurisy, lung abscess, and septicemia, as well as to cardiovascular disease [4, 5]. The risks of pneumonia and pneumonia-related mortality increase with age [6]; thus, the incidence of pneumonia is projected to increase as a result of global population aging [7].

Air pollution has emerged as a significant public health problem worldwide [8]. Despite plausible hypotheses linking air pollution with pneumonia [9–12], limited evidence of the association is available, especially in adults. Some studies have started to report some evidence of an association between ambient particulate matter (PM) pollution and respiratory infections, including pneumonia. A time series study suggested an association between fine particulate matter ($PM_{2.5}$, PM ≤2.5 μm in aerodynamic diameter) and increased admissions for respiratory tract infections in Medicare enrollees (aged >65 years) in 204 United States urban counties [13]. Similarly, another study reported significant effects of $PM_{10}$ (PM ≤10 μm in aerodynamic diameter) on pneumonia admissions among 36 US citizens aged 65 years or older [14]. Strosnider and colleagues assessed age-specific acute effects of $PM_{2.5}$ on respiratory emergency department visits in 17 US states and reported that short-term exposure to $PM_{2.5}$ was associated with increased emergency department visits for pneumonia in people aged 19–65 years, but not in people aged <19 or ≥65 years [15]. A recent case-crossover study from the US demonstrated significant associations between short-term elevations in $PM_{2.5}$ and greater healthcare utilization for acute lower respiratory infection in both children and adults [16]. Host and colleagues found increased admissions for lower respiratory tract infections following $PM_{2.5}$ exposure in six French cities [17], and two time series studies in Hong Kong provided further evidence on the acute effects of PM on respiratory tract infections/pneumonia [18, 19]. However, these studies were conducted primarily in high-income countries/cities where characteristics of air pollution and socioeconomic status differ from those in low- or middle-income countries. The effects of short-term exposure to PM pollution on pneumonia in low- or middle-income counties require further investigation.

China, the largest low- or middle-income country, has one of the highest $PM_{2.5}$ concentrations worldwide [20]. However, only a few studies have explored the associations between PM pollution and pneumonia, and were subjected to single city/hospital study and small sample size [21, 22]. In this study, we examined the short-term associations between concentrations of ambient PM pollution and daily hospital admissions for pneumonia in adults in China between 2014 and 2017.

## Methods

### Study sites

A total of 184 Chinese cities were finally included in this study. **Fig 1** shows the locations of the 184 cities, representing a geographic distribution across China. Cities were selected according to the availability of daily hospital admission and air pollution data. Cities with ≤1-year

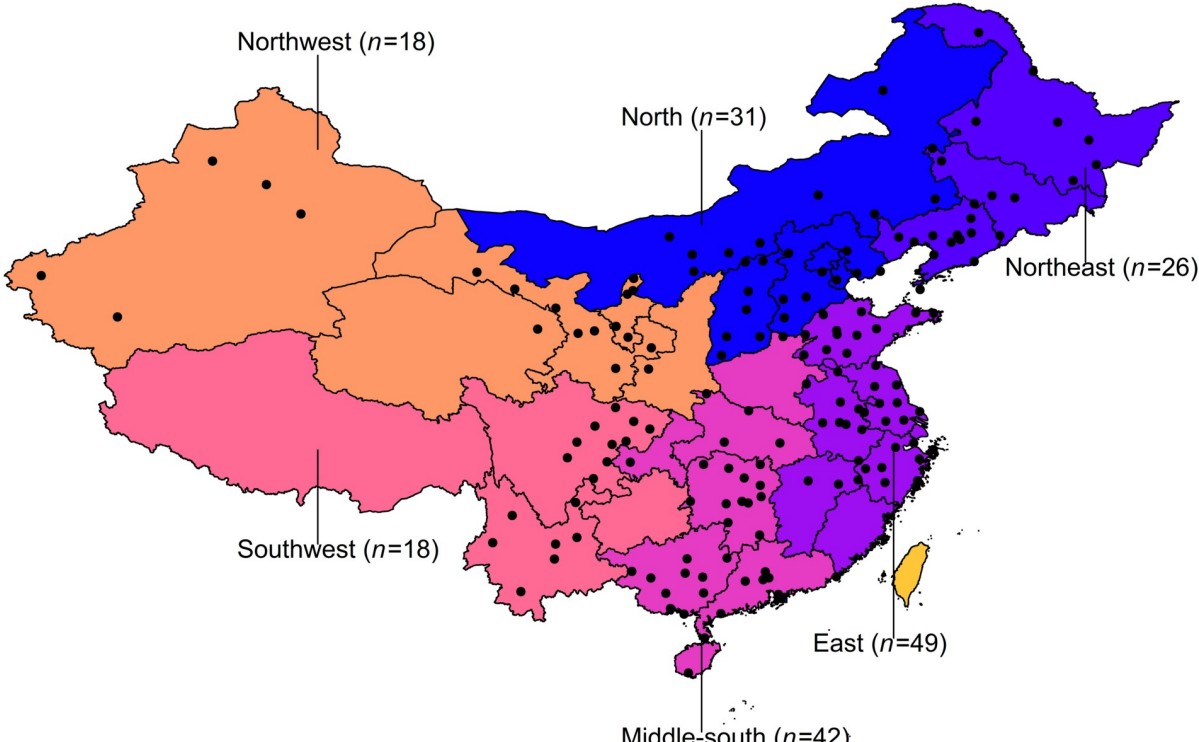

**Fig 1. Locations of the 184 Chinese cities included in the study.** *The base map was obtained from Natural Earth* (https://naturalearthdata.com).

hospital admission records were excluded given the fact that a long enough study period in a time series analysis is needed to ensure credible precision and power. Individuals' detailed disease diagnosis (International Classification of Diseases [ICD] code and text of disease diagnosis) was required to identify pneumonia admissions. Cities with no information on disease diagnosis recorded in the database were also excluded (e.g., Beijing and Shanghai). In addition, we applied additional rules to control the quality of the original data. We regressed the city-specific average numbers of pneumonia admission per day on the number of enrollers of the city. Cities with an absolute value of standardized residuals >3 were considered as the outliers and were removed from the analysis. China has gradually introduced $PM_{2.5}$ in the national air quality monitoring network and publicized real-time monitoring data since 2013. In this study, the total study period encompassed 2014–2017, and the years of analysis differed by city based on the availability of air pollution data. Of the 184 cities, 78 cities had 3-year PM data records and 106 cities had 4-year PM data records. The years of the study period for each city were presented in **S1 Table**.

## Study population

In China, there are three main health insurance schemes: the Urban Employee Basic Medical Insurance (UEBMI), the Urban Residence Basic Medical Insurance, and the New Rural Cooperative Medical Scheme, covering more than 92% of the population [23]. Private medical insurance covers little in China and is generally supplementary to the basic schemes. Daily pneumonia admission data were obtained from the UEBMI database. The UEBMI covers urban employees and retired employees (aged ≥18 years). All employers in urban areas, including government agencies and institutions, state-owned enterprises, private businesses,

social organizations, and other private entities and their employees (retirees included), are obligated to enroll in UEBMI [24]. At the end of 2016, 282.93 million individuals were recorded in the database. The original data source was medical claims data for urban employees, which included sex, age, date of admission, and cause of admission. Information regarding the number of individuals enrolled in the database, city residents, and coverage rates of these cities for the UEBMI database were published previously [25]. We have legal access to this national insurance information through the big data platform of national medical insurance data for risk prevention and control, established by the Ministry of Human Resources and Social Security of the People's Republic of China and Beijing HealthCom Data Technology. Hospitalizations with a principal diagnosis of pneumonia (ICD-10 codes J12–J18) were identified from the database. Because the health information was primarily the city-specific daily count of pneumonia admissions, i.e., summarized data (overall, and by age and sex subgroups) without any individual identifiers, this study was exempted from Institutional Review Board approval by the Ethics Committee of Peking University Health Science Center, Beijing, China. The need for informed consent was also waived by the Institutional Review Board. This study is reported as per the STROBE guideline (**S1 STROBE checklist**).

## Environmental data

Data on hourly $PM_{2.5}$ and $PM_{10}$ measurements in each city were obtained from the National Air Pollution Monitoring System. Each city has 1–17 monitoring stations. China has issued detailed standards on placement of monitoring stations and the air pollution monitoring process [26]; the monitoring stations cannot be located in the direct vicinity of apparent emission sources; thus, measurements reflect the general urban background concentrations. These measurements have therefore been used extensively to represent population exposure [27–29]. Data on sulfur dioxide ($SO_2$), nitrogen dioxide ($NO_2$), carbon monoxide (CO), and ozone ($O_3$) concentrations were also collected from the same source. We calculated 24-hour averages by averaging hourly air pollution measurements within a day. In each city, the measurements from all monitoring stations were averaged to represent population exposure [27, 28, 30]. Rates of missing data were 0.66% for $PM_{2.5}$ and 0.15% for $PM_{10}$. Days with missing monitoring measurements were excluded from the analysis. Data on hourly atmospheric temperature and relative humidity were provided by the China Meteorological Data Sharing Service System. Each city has 1–3 meteorological monitoring stations, all located in urban areas. Rates of missing temperature and relative humidity data were only 0.25%. Daily (24-hour) mean temperature and relative humidity were calculated by averaging all valid monitoring measurements in each city. More detailed information about health, air pollution, and meteorological data sources is given in our previous publications [24, 25].

## Statistical analysis

We used a common two-stage method to examine the associations between PM and pneumonia admissions [13, 24, 27, 30–32]. The method and the model used in this study were designed before the analyses were conducted, and the prespecified statistical analysis plan is present in **S1 Appendix**. In stage 1, we assessed the city-specific associations between PM and pneumonia admissions using a quasi-Poisson regression model that allows for over-dispersed admission counts. Several covariates were included in the model, following previous published studies [13, 27, 32, 33]: (1) a natural cubic spline for time with 7 degrees of freedom (df) per year to adjust for seasonality and long-term trends, (2) natural cubic splines for 3-day moving average temperature with 6 df and relative humidity with 3 df to control for nonlinear and delayed effects of meteorology, and (3) indicators of day of the week and public holiday. The

selection of df values was based on the parameter used in several recent large national studies [27, 28, 32, 33]. The model was as shown below:

$$Log[E(Y_t)] = \alpha + \beta(PM) + \text{day of the week} + \text{public holiday} + ns(\text{time}, df = 7/\text{year}) + ns(\text{temperature}, df = 6) + ns(\text{relative humidity}, df = 3)$$

where $E(Y_t)$ is the expected number of pneumonia admissions on day t; β indicates the log-relative risk of pneumonia associated with a unit increase of PM concentrations; $ns()$ indicates a natural cubic spline function; and temperature and relative humidity indicate 3-day moving averages. This model was in line with those used in previous studies [13, 24, 27, 30, 31] and was tested in sensitivity analyses, described in detail below. We also conducted unadjusted analyses with only PM and a natural cubic spline for time, with 7 df per year in the regression model. In stage 2, we conducted random-effects meta-analyses to combine the city-specific effect estimates. To explore regional heterogeneity of the associations, we divided the 184 cities into six geographical regions, i.e., East, Middle-south, Southwest, Northwest, North, and Northeast regions [27]. We used single lags of 0, 1, and 2 days to assess the temporal association between PM and pneumonia. Considering that single-day lag models may underestimate PM effects, we also estimated associations with 3-day (lag 0–2) moving average PM concentrations.

We plotted the national-average concentration-response curves of the associations of PM with pneumonia admissions using the approach of the Air Pollution and Health—A European Approach (APHEA) project [27, 32, 34]. Briefly, we replaced the linear term of PM in the main model with a B-spline function with two knots at 60 and 150 μg/m³ for $PM_{2.5}$ and 100 and 200 μg/m³ for $PM_{10}$, based on the distribution of PM concentrations in each city. We then estimated five regression coefficients of the spline function and the $5 \times 5$ variance-covariance matrix in each city. Finally, we fitted multivariable random-effect models to combine the city-specific components of spline estimates. Variance-covariance matrix is a matrix whose element in the $i, j$ position is the covariance between the $i$-th and $j$-th elements of a random vector. The regression coefficients derived from the main model with a B-spline function for PM.

We examined effect modification of the relationship between short-term ambient PM concentrations and pneumonia hospital admissions in analyses by sex, age, and region (northern and southern China). The regional division followed the Huai River-Qinling Mountains line [27, 30]. We assessed the differences in the estimates using a Z-test [35]. We also fitted meta-regression models to investigate several city characteristics as potential effect modifiers, including cities' annual-average air pollutants concentrations, air temperature and relative humidity, gross domestic product (GDP) per capita, and coverage rates by the UEBMI [27, 30]. Cities' annual-average air pollutants concentrations, temperature, and relative humidity were calculated from daily measurements during the study period. City-specific relative risks and their confidence intervals (CIs) as the outcome were meta-regressed on each continuous variable of city characteristics. We fitted the random-effects meta-regression model using city-level estimations from the first stage as $y_i = \beta x_i + \mu_i + e_i$, where $y_i$ is the city-specific estimation, $x_i$ is the city-level characteristic variable, $\mu_i$ is the city-specific random effect characterized by the between-city variance, and $e_i$ is a normal error term [36]. Random-effects meta-regression can be considered as an extension to the random-effects meta-analysis (i.e., DerSimonian and Laird model) that includes study-level (i.e., city-level in our case) covariates. The model is implemented by the metareg function in STATA. The algorithm for random-effects meta-regression firstly estimated the between-study (between-city) variance and then estimated the coefficients, β, by weighted least squares by using the weights $1/(\sigma_i^2 + \tau^2)$, where $\sigma_i^2$ is the

standard error of the estimated effect in study (city) i, and $\tau^2$ is the between-study (between-city) variance, advocated by Thompson and Sharp [37].

## Sensitivity analysis

We conducted several sensitivity analyses: (1) We tested the potential confounding effects of gaseous pollutants in two-pollutant models; (2) we separately examined the associations of PM with pneumonia in cities with only 3- or 4-year data to compare with the primary analysis; (3) we examined the associations between PM and pneumonia in cities with different levels of population coverage by the UEBMI (<20% and ≥20%); (4) we examined the associations excluding cities with ≤2 monitoring stations; (5) we conducted a sensitivity analysis with additional adjustment of the hospitalization of influenza; and (6) to examine whether we appropriately specified the regression model used in this analysis, we used alternative df for time (6–10 per year) or used penalized spline functions for time and meteorological variables in the model.

Statistical analyses were conducted in R version 3.2.2 (R Development Core Team 2008) and STATA software, version 12 (StataCorp, College Station, TX), and a two-sided $P < 0.05$ was considered statistically significant. The effect estimates are presented as percentage increases and 95% CIs in pneumonia admissions per 10-μg/m$^3$ increase in PM concentrations.

## Results

There were 4.2 million hospital admissions for pneumonia in the 184 Chinese cities between 2014 and 2017 in our study. Table 1 presents summary statistics on citywide pneumonia admissions, PM concentrations, and weather conditions at an annual-average level during the study period. The mean number of daily hospital admissions for pneumonia was 26 (range: 1 to 376). The national-average PM$_{2.5}$ concentration was 50 μg/m$^3$ (range: 15 to 102 μg/m$^3$), and the mean PM$_{10}$ concentration was 89 μg/m$^3$ (range: 28 to 193 μg/m$^3$). The average citywide annual-mean temperature was 14°C (range: 0 to 24°C). City-specific characteristics, including the number of individuals enrolled in the database, city residents, coverage rates of the population by UEBMI database, and annual-average PM$_{2.5}$ and PM$_{10}$ concentrations, temperature, relative humidity, and daily pneumonia hospitalizations are presented in S1 Table.

Table 2 shows the Spearman correlation coefficient values for the environmental variables. At the national level, daily PM$_{2.5}$ and PM$_{10}$ concentrations were positively correlated with SO$_2$ (PM$_{2.5}$: r = 0.56, $P < 0.001$; PM$_{10}$: r = 0.59, $P < 0.001$), NO$_2$ (PM$_{2.5}$: r = 0.64, $P < 0.001$; PM$_{10}$: r = 0.64, $P < 0.001$), and CO (PM$_{2.5}$: r = 0.61, $P < 0.001$; PM$_{10}$: r = 0.57, $P < 0.001$) concentrations. O$_3$ concentrations were not correlated with PM$_{2.5}$ (r = −0.02, $P = 0.452$) or PM$_{10}$ (r = 0.05, $P = 0.215$). There were inverse and weak correlations between PM$_{2.5}$ and PM$_{10}$ concentrations and temperature (PM$_{2.5}$: r = −0.26, $P < 0.001$; PM$_{10}$: r = −0.26, $P < 0.001$) and relative humidity (PM$_{2.5}$: r = −0.08, $P < 0.001$; PM$_{10}$: r = −0.28, $P < 0.001$).

City-specific estimates of the associations between 3-day moving average (lag 0–2) concentrations of PM$_{2.5}$ and PM$_{10}$ and hospital admissions for pneumonia are presented in S2 Table. Table 3 shows the national-average percentage increases in daily hospital admissions for pneumonia associated with a 10-μg/m$^3$ increase in PM$_{2.5}$ and PM$_{10}$ concentrations at lags 0, 1, 2, and 0–2 days. Overall, we observed similar lag patterns for PM$_{2.5}$ and PM$_{10}$. For single-day lags, the strongest associations occurred at lag 0. The associations weakened considerably at lag days 1 and 2. Lag 0–2 generated the highest estimates. On average, a 10-μg/m$^3$ increase in PM$_{2.5}$ and PM$_{10}$ concentrations at lag 0–2 corresponded to a 0.31% (0.15% to 0.46%, $P < 0.001$) and 0.19% (0.11% to 0.30%, $P < 0.001$) increase in pneumonia admissions, respectively.

**Table 1. Summary statistics of citywide annual-mean hospital admissions for pneumonia, air pollutants, and weather conditions in 184 Chinese cities, 2014–2017.**

| Variables | Mean ± SD | Minimum | Percentile | | | Maximum |
|---|---|---|---|---|---|---|
| | | | 25th | 50th | 75th | |
| Daily pneumonia admissions | 26 ± 58 | 1 | 5 | 9 | 19 | 376 |
| $PM_{2.5}$ (μg/m³) | 50 ± 19 | 15 | 38 | 49 | 59 | 102 |
| $PM_{10}$ (μg/m³) | 89 ± 40 | 28 | 66 | 84 | 103 | 193 |
| Temperature (˚C) | 14 ± 5 | 0 | 11 | 15 | 18 | 24 |
| Relative humidity (%) | 68 ± 12 | 34 | 59 | 70 | 78 | 92 |

Abbreviations: $PM_{2.5}$, particulate matter ≤2.5 μm in aerodynamic diameter; $PM_{10}$, particulate matter ≤10 μm in aerodynamic diameter

**Table 4** presents the results of stratified analyses. The estimates were similar in males ($PM_{2.5}$: percentage increase, 0.31%; 95% CI 0.12%–0.49%; $PM_{10}$: percentage increase, 0.20%; 95% CI 0.13%–0.28%) and females ($PM_{2.5}$: percentage increase, 0.33%; 95% CI 0.13%–0.53%; $PM_{10}$: percentage increase, 0.15%; 95% CI 0.05%–0.25%) (the *P* values for the differences in the estimates were 0.887 for $PM_{2.5}$ and 0.422 for $PM_{10}$). The associations between PM and pneumonia were stronger in people aged ≥75 years, but the difference between the strata was only significant for $PM_{10}$ (the *P* values for the differences in the estimates were 0.201 for $PM_{2.5}$ and *P* < 0.001 for $PM_{10}$). The effect estimates were higher in the southern region ($PM_{2.5}$: percentage increase, 0.40%; 95% CI 0.16%–0.65%; $PM_{10}$: percentage increase, 0.35%; 95% CI 0.16%–0.54%) than in the northern region ($PM_{2.5}$: percentage increase, 0.23%; 95% CI 0.04%–0.43%; $PM_{10}$: percentage increase, 0.12%; 95% CI 0%–0.24%), but the between-region difference was not significant for $PM_{2.5}$ (the *P* values for the regional differences in the estimates were 0.276 for $PM_{2.5}$ and 0.045 for $PM_{10}$). We grouped the cities further into six geographical areas; the average estimates for the six areas are presented in **S3 Table**. There was a significant heterogeneity in the PM–hospitalization associations across different regions. The associations were more evident in the Middle-south, East, and North regions.

We noted a slightly nonlinear concentration-response curve between $PM_{2.5}$ and pneumonia admissions, where there appears to be a plateau at higher concentrations. This evidence of linearity is consistent with studies that examined short-term PM exposure and other outcomes [13–15, 19, 27]. The curve of the association between $PM_{10}$ and pneumonia increased sharply at concentrations below 100 μg/m³ and then climbed relatively moderately as concentrations

**Table 2. Spearman correlation coefficients among the environmental variables in 184 Chinese cities, 2014–2017.**

| Variables | $PM_{2.5}$ | $PM_{10}$ | $SO_2$ | $NO_2$ | CO | $O_3$ | Temp | RH |
|---|---|---|---|---|---|---|---|---|
| $PM_{2.5}$ | 1.00 | 0.90* | 0.56* | 0.64* | 0.61* | −0.02 (P = 0.452) | −0.26* | −0.08* |
| $PM_{10}$ | — | 1.00 | 0.59* | 0.64* | 0.57* | 0.05 (P = 0.215) | −0.26* | −0.28* |
| $SO_2$ | — | — | 1.00 | 0.55* | 0.52* | −0.09* | −0.38* | −0.33* |
| $NO_2$ | — | — | — | 1.00 | 0.54* | −0.13* | −0.31* | −0.12* |
| CO | — | — | — | — | 1.00 | −0.21* | −0.28* | −0.04* |
| $O_3$ | — | — | — | — | — | 1.00 | 0.53* | −0.23* |
| Temp | — | — | — | — | — | — | 1.00 | 0.26* |
| RH | — | — | — | — | — | — | — | 1.00 |

*P < 0.001.

Abbreviations: $PM_{2.5}$, particulate matter ≤2.5 μm in aerodynamic diameter; $PM_{10}$, particulate matter ≤10 μm in aerodynamic diameter; RH, relative humidity; Temp, temperature

**Table 3. National-average percentage increase with 95% CI in daily hospital admissions for pneumonia associated with a 10 μg/m³ increase in PM$_{2.5}$ and PM$_{10}$ concentrations using different lag days in 184 Chinese cities, 2014–2017.**

| Lag day | Percentage increase (95% CI) | P value |
|---|---|---|
| Unadjusted analyses | | |
| PM$_{2.5}$ | | |
| Lag 0 | 0.14 (0.04–0.24) | 0.007 |
| Lag 1 | 0.11 (0.01–0.21) | 0.027 |
| Lag 2 | 0.15 (0.06–0.25) | 0.002 |
| Lag 0–2 | 0.23 (0.08–0.37) | 0.002 |
| PM$_{10}$ | | |
| Lag 0 | 0.11 (0.04–0.18) | 0.001 |
| Lag 1 | 0.06 (−0.01–0.13) | 0.093 |
| Lag 2 | 0.10 (0.02–0.17) | 0.008 |
| Lag 0–2 | 0.15 (0.04–0.25) | 0.005 |
| Adjusted analyses* | | |
| PM$_{2.5}$ | | |
| Lag 0 | 0.23 (0.14–0.33) | <0.001 |
| Lag 1 | 0.15 (0.05–0.25) | 0.004 |
| Lag 2 | 0.11 (0.01–0.21) | 0.039 |
| Lag 0–2 | 0.31 (0.15–0.46) | <0.001 |
| PM$_{10}$ | | |
| Lag 0 | 0.18 (0.12–0.25) | <0.001 |
| Lag 1 | 0.08 (0.02–0.15) | 0.015 |
| Lag 2 | 0.04 (−0.02–0.11) | 0.218 |
| Lag 0–2 | 0.19 (0.11–0.30) | <0.001 |

*Adjusted for temperature, relative humidity, calendar time, day of the week, and public holiday.

Abbreviations: PM$_{2.5}$, particulate matter ≤2.5 μm in aerodynamic diameter; PM$_{10}$, particulate matter ≤10 μm in aerodynamic diameter

increased (Fig 2). The estimated risk of PM$_{10}$ concentrations for pneumonia was 0.38% (0.18% to 0.57%, $P < 0.001$) at PM$_{10} < 100$ μg/m³ and 0.10% (−0.10% to 0.30%, $P = 0.754$) at PM$_{10} \geq 100$ μg/m³.

Table 5 summarizes the results of meta-regression analyses of effect modification on the associations between PM and pneumonia by city-level characteristics. We observed stronger associations between PM$_{10}$ and pneumonia in cities with higher annual-average temperatures or relative humidity. For each 10-μg/m³ increase in PM$_{10}$ concentrations, a city with 1°C higher annual-mean temperature with respect to another city would see an additional 0.031% (0.003% to 0.058%, $P = 0.026$) increase in pneumonia admissions; a city with 1% higher annual-mean relative humidity with respect to another city would see an additional 0.011% (0% to 0.022%, $P = 0.045$) increase in pneumonia admissions. We observed the same direction of effect modification by temperature ($P = 0.129$) and relative humidity ($P = 0.378$) on the association between PM$_{2.5}$ and pneumonia, but the effects were not significant. No evidence was found for effect modification by annual-average PM$_{2.5}$ (percentage increase, −0.019%; 95% CI −0.113% to 0.075%, $P = 0.688$) or PM$_{10}$ (percentage increase, 0; 95% CI −0.064% to 0.064%, $P = 0.997$). Annual-average NO$_2$ (the $P$ values were 0.737 for PM$_{2.5}$ and $P = 0.861$ for PM$_{10}$) or CO (the $P$ values were 0.466 for PM$_{2.5}$ and $P = 0.367$ for PM$_{10}$) concentrations, GDP per capita (the $P$ values were 0.864 for PM$_{2.5}$ and $P = 0.655$ for PM$_{10}$), or the coverage rate of

**Table 4. National-average percentage increase (PI) with 95% CI in daily hospital admissions for pneumonia associated with a 10 μg/m$^3$ increase in PM$_{2.5}$ and PM$_{10}$ concentrations (lag 0–2), stratified by sex, age, and geographical region.**

| Variables | PM$_{2.5}$ | | PM$_{10}$ | |
|---|---|---|---|---|
| | PI (95% CI) | P value$^#$ | PI (95% CI) | P value$^#$ |
| **Unadjusted analyses** | | | | |
| Sex | | 0.401 | | 0.279 |
| Male | 0.26 (0.10–0.41), P = 0.001 | | 0.18 (0.07–0.28), P = 0.001 | |
| Female | 0.16 (−0.01–0.33), P = 0.074 | | 0.09 (−0.03–0.21), P = 0.153 | |
| Age, years | | | | |
| 18–64 | 0.04 (−0.12–0.21), P = 0.605 | 1 (Ref.) | 0.01 (−0.11–0.13), P = 0.870 | 1 (Ref.) |
| 65–74 | 0.28 (0.10–0.46), P = 0.003 | 0.051 | 0.18 (0.05–0.30), P = 0.006 | 0.060 |
| ≥75 | 0.46 (0.29–0.63), P < 0.001 | <0.001 | 0.33 (0.21–0.45), P < 0.001 | <0.001 |
| Region | | 0.007 | | <0.001 |
| South | 0.46 (0.21–0.72), P < 0.001 | | 0.45 (0.27–0.63), P < 0.001 | |
| North | 0.05 (−0.11–0.22), P = 0.529 | | −0.06 (−0.17–0.05), P = 0.313 | |
| **Adjusted analyses**$^*$ | | | | |
| Sex | | 0.887 | | 0.422 |
| Male | 0.31 (0.12–0.49), P < 0.001 | | 0.20 (0.13–0.28), P < 0.001 | |
| Female | 0.33 (0.13–0.53), P < 0.001 | | 0.15 (0.05–0.25), P = 0.001 | |
| Age, years | | | | |
| 18–64 | 0.24 (0.04–0.44), P = 0.016 | 1 (Ref.) | 0.10 (0.02–0.19), P = 0.017 | 1 (Ref.) |
| 65–74 | 0.35 (0.12–0.58), P < 0.001 | 0.479 | 0.20 (0.10–0.30), P < 0.001 | 0.126 |
| ≥75 | 0.42 (0.23–0.62), P < 0.001 | 0.201 | 0.32 (0.22–0.39), P < 0.001 | <0.001 |
| Region | | 0.276 | | 0.045 |
| South | 0.40 (0.16–0.65), P < 0.001 | | 0.35 (0.16–0.54), P < 0.001 | |
| North | 0.23 (0.04–0.43), P = 0.022 | | 0.12 (0–0.24), P = 0.046 | |

$^*$Adjusted for temperature, relative humidity, calendar time, day of the week, and public holiday.

$^#$P value obtained from Z-test for the difference between the two risk estimates derived from subgroup analysis.

Abbreviations: PM$_{2.5}$, particulate matter ≤2.5 μm in aerodynamic diameter; PM$_{10}$, particulate matter ≤10 μm in aerodynamic diameter; Ref., reference

the population (the P values were 0.835 for PM$_{2.5}$ and P = 0.665 for PM$_{10}$) did not significantly modify the associations.

The associations of PM$_{2.5}$ and PM$_{10}$ with pneumonia remained after adjusting for SO$_2$ (PM$_{2.5}$: percentage increase, 0.22%; 95% CI 0.07%–0.37%; P = 0.006; PM$_{10}$: percentage increase, 0.12%; 95% CI 0.02%–0.22%; P = 0.015) and O$_3$ (PM$_{2.5}$: percentage increase, 0.31%; 95% CI 0.16%–0.46%; P < 0.001; PM$_{10}$: percentage increase, 0.20%; 95% CI 0.09%–0.30%; P < 0.001); the effects remained positive but not significant after controlling for CO (PM$_{2.5}$: percentage increase, 0.12%; 95% CI −0.03%–0.27%; P = 0.127; PM$_{10}$: percentage increase, 0.06%; 95% CI −0.04%–0.16%; P = 0.256) and NO$_2$ (PM$_{2.5}$: percentage increase, 0.10%; 95% CI −0.05%–0.25%; P = 0.271; PM$_{10}$: percentage increase, 0.04%; 95% CI −0.06%–0.14%; P = 0.431) (S4 Table).

S5 Table lists the results of the sensitivity analyses. Consistent associations were observed in datasets with different lengths of data-years. Specifically, for PM$_{2.5}$, the increases in pneumonia admissions were 0.28% (0% to 0.56%, P = 0.044) in 78 cities with 3-year data and 0.37% (0.20% to 0.56%, P < 0.001) in 106 cities with 4-year data; for PM$_{10}$, the corresponding values were 0.13% (−0.07% to 0.34%, P = 0.213) in cities with 3-year data and 0.24% (0.11% to 0.37%, P < 0.001) in cities with 4-year data. The estimates were similar in cities with different levels of population coverage (<20% and ≥20%). The estimates were slightly attenuated after excluding

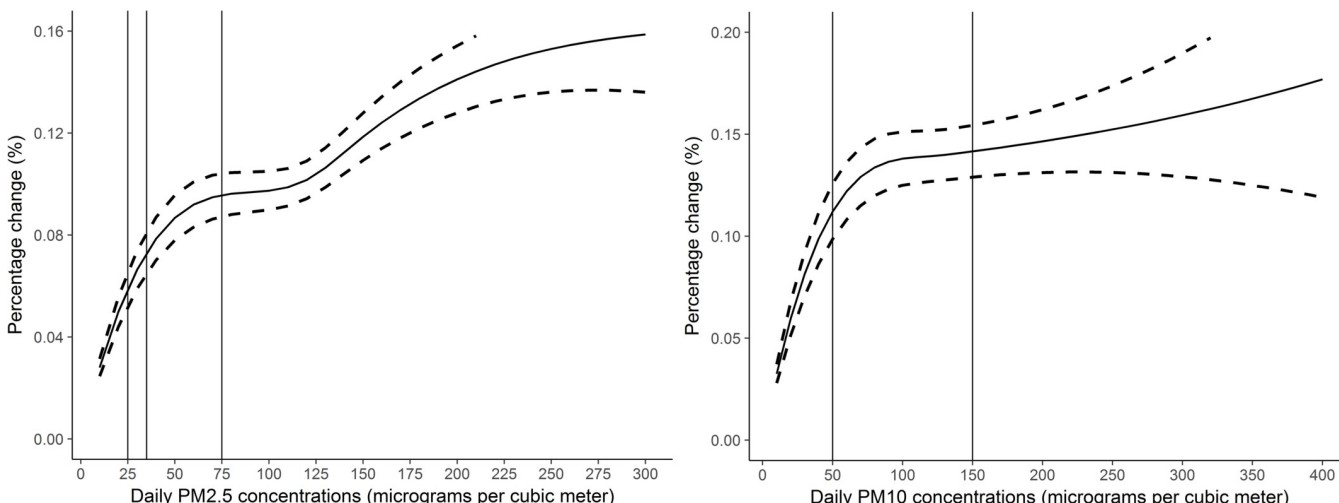

**Fig 2. National-average concentration-response curves of 3-day moving average (lag 0–2) concentrations of PM$_{2.5}$ and PM$_{10}$ and hospital admissions for pneumonia in 184 cities in China, 2014–2017.** Note, y-axes of the two graphs are scaled differently. The solid line represents relative changes and the dashed lines represent the 95% CIs. The vertical lines represent the air quality guidelines or standards for 24-hour average concentrations of PM$_{2.5}$ and PM$_{10}$. PM$_{2.5}$: 25 μg/m$^3$ is the World Health Organization (WHO) air quality guideline for daily PM$_{2.5}$ concentrations, 35 μg/m$^3$ is the Chinese Grade I standard for daily PM$_{2.5}$ concentrations, and 75 μg/m$^3$ is the Chinese Grade II standard; PM$_{10}$: 50 μg/m$^3$ is both the WHO air quality guideline and Chinese Grade I standard for daily PM$_{10}$ concentrations, and 150 μg/m$^3$ is the Chinese Grade II standard for daily PM$_{10}$ concentrations. PM$_{2.5}$, particulate matter ≤2.5 μm in aerodynamic diameter; PM$_{10}$, particulate matter ≤10 μm in aerodynamic diameter.

cities with ≤2 monitoring stations (PM$_{2.5}$: percentage increase, 0.30%; 95% CI 0.15%–0.46%, $P < 0.001$; PM$_{10}$: percentage increase, 0.21%; 95% CI 0.10%–0.32%, $P < 0.001$) or after adjustment of hospitalization of influenza (PM$_{2.5}$: percentage increase, 0.29%; 95% CI 0.13%–0.45%,

**Table 5. Meta-regression results of the modification effects of city-level characteristics on the associations between PM$_{2.5}$ and PM$_{10}$ and hospital admissions for pneumonia in 184 Chinese cities, 2014–2017.**

| City-level characteristics | Percentage increase | 95% CI | P value |
|---|---|---|---|
| PM$_{2.5}$ | | | |
| PM$_{2.5}$ (10 μg/m$^3$) | −0.019 | −0.113–0.075 | 0.688 |
| NO$_2$ (10 μg/m$^3$) | −0.030 | −0.208–0.148 | 0.737 |
| CO (1 mg/m$^3$) | 0.177 | −0.300–0.657 | 0.466 |
| Temperature (°C) | 0.031 | −0.009–0.071 | 0.129 |
| Relative humidity (%) | 0.007 | −0.009–0.023 | 0.378 |
| GDP per capita (¥10,000) | 0.007 | −0.073–0.087 | 0.864 |
| Coverage rate (%) | −0.010 | −0.106–0.087 | 0.835 |
| PM$_{10}$ | | | |
| PM$_{10}$ (10 μg/m$^3$) | 0 | −0.064–0.064 | 0.997 |
| NO$_2$ (10 μg/m$^3$) | −0.011 | −0.135–0.113 | 0.861 |
| CO (1 mg/m$^3$) | 0.153 | −0.180–0.488 | 0.367 |
| Temperature (°C) | 0.031 | 0.003–0.058 | 0.026 |
| Relative humidity (%) | 0.011 | 0–0.022 | 0.045 |
| GDP per capita (¥10,000*) | −0.012 | −0.066–0.042 | 0.655 |
| Coverage rate (%) | −0.002 | −0.008–0.005 | 0.665 |

*¥10,000 = £1,169; $1,456; 1,377.

Abbreviations: GDP, gross domestic product; PM$_{2.5}$, particulate matter ≤2.5 μm in aerodynamic diameter; PM$_{10}$, particulate matter ≤10 μm in aerodynamic diameter

$P < 0.001$; $PM_{10}$: percentage increase, 0.18%; 95% CI 0.07%–0.29%, $P < 0.001$). The national-average estimates generated from models with alternative df for time (6–10 per year, all $P < 0.05$) or with penalized spline functions for time and meteorological variables ($PM_{2.5}$: percentage increase, 0.31%; 95% CI 0.17%–0.45%, $P < 0.001$; $PM_{10}$: percentage increase, 0.22%; 95% CI 0.12%–0.32%, $P < 0.001$) were comparable to those generated from the base model.

## Discussion

Using comprehensive data, we examined the associations between ambient PM pollution and pneumonia hospitalizations in 184 Chinese cities. We also investigated potential effect modifications by a variety of demographic, geographical, and meteorological characteristics. We found that short-term elevations in $PM_{2.5}$ and $PM_{10}$ were associated with higher pneumonia-related hospital admissions in Chinese adults. To our knowledge, this is the first study in China, or even in the low- or middle-income countries, to investigate whether short-term changes in ambient PM concentrations are related to increases pneumonia hospital admissions on a national scale.

We observed increased risk of hospital admissions for pneumonia in association with both $PM_{2.5}$ and $PM_{10}$ at lag 0 and lag 1, in line with previous findings from studies examining short-term PM exposures [14, 21]. The highest estimates for single-day lag models were observed for lag 0, indicating immediate PM effects on pneumonia. In addition, the estimate at lag 0–2 was slightly higher than at the estimate for lag 0, suggesting that there might be some cumulative effects of PM on pneumonia admissions as well. We also explored the concentration-response curve of the associations between PM and pneumonia hospitalizations. China is one of the most polluted countries worldwide [38]. The average city-specific annual-mean $PM_{2.5}$ and $PM_{10}$ concentrations during the study period were 50 μg/m$^3$ and 89 μg/m$^3$, respectively, allowing us to assess the concentration-response pattern at high concentrations. The curves plateau at high concentrations, indicating that the increase in risk of pneumonia admission is greater at lower compared with higher concentrations. A similar shape of the association was also observed in other studies [27]. The leveling off at high concentrations might be explained in that people vulnerable to PM exposure may have developed symptoms and sought treatment before PM concentrations reached high concentrations. Other reasons could also be that there may be different health risks associated with different PM sizes and that people avoid spending time outdoors or start wearing face masks when air pollution is severe.

The $PM_{2.5}$ estimates were higher than those of $PM_{10}$ at all lag days in this study, in line with previous findings on pneumonia [22, 39]. $PM_{10}$ is representing exposure to more particles in the coarse range ($PM_{10–2.5}$) than $PM_{2.5}$. The sources, composition, and lung deposition patterns of $PM_{10–2.5}$ vary from those of $PM_{2.5}$ [40, 41]. The chemical composition of PM differs by size, with more crustal materials in $PM_{10–2.5}$ and more combustion-related constituents in $PM_{2.5}$ [31, 41]. It was reported that $PM_{2.5}$ can penetrate deep into the lungs, reaching the bronchioles and depositing inside the alveoli. In addition, $PM_{2.5}$ has a larger surface area than $PM_{10}$ and thus can absorb more toxic substances per unit mass [42].

There are several possible mechanisms linking PM pollution to pneumonia. It has been postulated that air pollution may act as an irritant and evoke defensive responses in the airways, including increased mucus secretion and bronchial hyperreactivity [9]. PM is a potent oxidant that can produce free radicals and cause oxidative stress in lung cells [10]. An animal study demonstrated that short-term exposure to concentrated ambient particles could increase the levels of reactive oxygen species in the lung [11]. PM-induced oxidative stress might impair the cellular defense and immune system, increasing the susceptibility to infection [19]. In addition, PM might enhance the susceptibility to infection through the suppression of the immune

response [43].These toxicological evidences were consistent with our findings on the short-term association between ambient PM concentrations and pneumonia hospitalizations. However, it should be noted that our study does not reveal the mechanisms related to PM that may be active in these processes, and thus further study is still required to verify the causality.

In the two-pollutant models, the associations between $PM_{2.5}$ and $PM_{10}$ and pneumonia remained positive but were not significant after adjusting for CO and $NO_2$ concentrations. Similarly, a case-crossover study by Tsai and Yang demonstrated a positive association between $PM_{2.5}$ and hospital admissions for pneumonia in a single-pollutant model, but the estimate decreased and became nonsignificant when controlling for CO and $NO_2$ [44]. We previously reported positive effects of $PM_{2.5}$ and $PM_{10}$ on hospital admissions for respiratory disease, but the estimates decreased dramatically and even became negative after adjusting for $NO_2$ [45]. $NO_2$ and CO are closely associated with vehicle exhaust emissions [46, 47]. In China, a substantial fraction of $PM_{2.5}$ originates in traffic emissions [48]. The high correlation between pollutants in the regression model may have weakened the observed effects, although the CIs of the estimates were not remarkably inflated in the two-pollutant model. Another possible explanation is that the effects of PM may be confounded by those of other air pollutants. Additional studies are warranted to investigate the independent effects of PM on pneumonia.

There were some effect modifiers in the PM-pneumonia associations. First, the elderly had a higher risk of PM-associated pneumonia. Biological functions decline in the elderly, and this population has a higher prevalence of chronic health conditions [9, 49] that may contribute to the increased vulnerability to air pollution. Second, the effects were stronger in cities with higher temperatures (or higher relative humidity, as they are highly correlated). There are several possible explanations. High temperature itself is associated with increased risk of pneumonia [50, 51]. It was reported that high temperature impacts the emission, transportation, dilution, chemical transformation, and deposition of pollutants [52]. Thus, there may exist the interaction between high temperatures and PM. Previous studies also show moderate evidence of the modifying effect of temperature on $PM_{10}$. A meta-analysis reported that the relative risks for respiratory mortality per 10 μg/m$^3$ increase in $PM_{10}$ in the low, middle, and high temperature level were 1.005 (95% CI 1.000–1.010), 1.008 (1.006–1.010), and 1.019 (1.010–1.028), respectively [53]. This is consistent with our findings. In addition, the larger estimate might be associated with exposure patterns. People generally spend more time outdoors in warmer cities, resulting in smaller measurement errors [52, 54].

This study provided detailed estimates of the risk of PM-associated hospital admission for pneumonia through the use of data from a national health insurance program that serves approximately one fifth of the population in China, and offered a unique advantage in identifying potential modifiers of the associations. Our study also has some limitations. First, the use of citywide monitoring measurements as a proxy for population exposure could cause exposure measurement errors, which tend to bias the estimates downward [55]. Second, children (aged <18 years) were not included in this study because UEBMI is for urban working and retired employees, although young children may be more sensitive to air pollution, similarly to the elderly. In addition, only urban employed and retired individuals were included in this study based on the type of insurance enrolled. Due to the differences in sociodemographic characteristics and PM concentrations between rural and urban areas, the generalizability of our findings should be interpreted with caution. Third, both one-stage and two-stage methods could be applied in multicity analysis. In this study, the two-stage method was applied, considering the issue of the computational complexity [56], in line with previous studies [13, 27, 32]. The variances estimated in the first step in the two-stage analysis may not be exactly those in the one-stage method. However, it was reported that one-stage and two-stage methods often generate very similar results [57]. Fourth, in this study, we examined the associations between

PM and hospital admissions for pneumonia, not including emergency room visits. As hospital admissions generally include the more severe cases, future investigations are needed to assess the effects of PM on other morbidity outcomes (e.g., emergency room visits). Fifth, as in other environmental health studies using a large administrative health database [13, 27, 30, 58], data on several patient-level variables, such as medical history and smoking status, were not available, limiting the ability to identify potentially susceptible populations. More detailed individual information would be needed to assess the modifying effects of individual characteristics on the associations between PM pollution and pneumonia.

In summary, we found that short-term elevations in PM were associated with increased pneumonia-related hospital admissions in Chinese adults. Our findings support the rationale to further limit PM concentrations in China.

## Supporting information

**S1 Table. Summary statistics on city-level characteristics in 184 Chinese cities.**
(DOCX)

**S2 Table. City-specific percentage increase with 95% CI in daily hospital admissions for pneumonia associated with a 10 μg/m$^3$ increase in PM$_{2.5}$ and PM$_{10}$ concentrations (lag 0–2) in 184 Chinese cities, 2014–2017.**
(DOCX)

**S3 Table. Regional-average percentage increase with 95% CI in daily hospital admissions for pneumonia associated with a 10 μg/m$^3$ increase in PM$_{2.5}$ and PM$_{10}$ concentrations (lag 0–2) in 184 Chinese cities, 2014–2017.**
(DOCX)

**S4 Table. National-average percentage increase with 95% CI in daily hospital admissions for pneumonia associated with a 10 μg/m$^3$ increase in PM$_{2.5}$ and PM$_{10}$ concentrations (lag 0–2) in two-pollutant models in 184 Chinese cities, 2014–2017.**
(DOCX)

**S5 Table. Results of sensitivity analyses.**
(DOCX)

**S1 STROBE checklist.**
(DOC)

**S1 Appendix. Statistical analysis plan.**
(DOCX)

## Author Contributions

**Conceptualization:** Yaohua Tian, Pei Gao, Yonghua Hu.

**Data curation:** Yaqin Si, Libo Chen, Chen Wei.

**Formal analysis:** Yaohua Tian, Hui Liu, Man Li, Tao Wu, Pei Gao.

**Funding acquisition:** Pei Gao, Yonghua Hu.

**Investigation:** Yaqin Si, Libo Chen, Chen Wei.

**Methodology:** Yaohua Tian, Mengying Wang, Pei Gao.

**Project administration:** Pei Gao.

**Resources:** Yaqin Si, Libo Chen, Chen Wei.

**Software:** Yaohua Tian, Yao Wu, Pei Gao.

**Supervision:** Chen Wei, Pei Gao, Yonghua Hu.

**Validation:** Yaohua Tian, Yiqun Wu, Xiaowen Wang, Pei Gao.

**Visualization:** Yaohua Tian, Hui Liu, Pei Gao.

**Writing – original draft:** Yaohua Tian, Yonghua Hu.

**Writing – review & editing:** Yaohua Tian, Hui Liu, Yiqun Wu, Yaqin Si, Man Li, Yao Wu, Xiaowen Wang, Mengying Wang, Libo Chen, Chen Wei, Tao Wu, Pei Gao, Yonghua Hu.

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
