## [Decision Letter · Decision Letter 0]

9 Oct 2019

Dear Dr. Hu,

Thank you very much for submitting your manuscript "Ambient Particulate Matter Pollution and Adult Hospital Admissions for Pneumonia in Urban China: A national time-series analysis" (PMEDICINE-D-19-02197) for consideration at PLOS Medicine. 

Your paper was evaluated by a senior editor and discussed among all the editors here. It was also discussed with an academic editor with relevant expertise, and sent to four independent reviewers, including a statistical reviewer. The reviews are appended at the bottom of this email and any accompanying reviewer attachments can be seen via the link below:

[LINK]

In light of these reviews, I am afraid that we will not be able to accept the manuscript for publication in the journal in its current form, but we would like to consider a revised version that addresses the reviewers' and editors' comments. Obviously we cannot make any decision about publication until we have seen the revised manuscript and your response, and we plan to seek re-review by one or more of the reviewers. 

We expect to receive your revised manuscript by Oct 30 2019 11:59PM. Please email us (plosmedicine@plos.org) if you have any questions or concerns.

We look forward to receiving your revised manuscript. 

Sincerely,

Caitlin Moyer, Ph.D.

Associate Editor 

PLOS Medicine

plosmedicine.org

1.Did your study have a prospective protocol or analysis plan? Please state this (either way) early in the Methods section.

c) In either case, changes in the analysis—including those made in response to peer review comments—should be identified as such in the Methods section of the paper, with rationale.

2. Abstract Methods and Findings: Line 29: Please change 0.28 billion adults to a more intuitive number (perhaps report in millions instead of billions).

3. Abstract Methods and Findings: Please include the study design.

4. Abstract Methods and Findings: Please quantify the results of the association between PM and hospital admissions with p values.

5. Abstract Methods and Findings: Please quantify the results of the effects of PM with higher temperatures/humidity and in elderly adults with 95% CIs and p values.

6. Abstract Conclusions: Please avoid assertions of primacy (“As the first study in China to investigate…”).

7. Abstract Conclusions: Your study is observational and therefore causality cannot be inferred. Please remove language that implies causality (...may assist in understanding how PM causes lung-inflammation diseases.).

9. Methods: Please provide details on the 184 cities included in the study, including criteria for the selection of cities and the years of the study period for each city.

10. Results: Line 200: Please provide a range for the number of daily hospital admissions for pneumonia.

11. Results: Line 268: Please change “statistically insignificant” to “effects were not significant after controlling for…”

12. Results: Lines 270-277: Please quantify these results with 95% CIs and P values, and clarify that the results are presented in S4 Table.

13. Discussion: Lines 280-285: Your study is observational and thus causality cannot be inferred. Please revise throughout and remove causal language (such as, “...we assessed the effects of ambient PM pollution on pneumonia admissions…” and “...investigate the acute effects of PM on pneumonia…”).

14. Discussion: Lines 289-290: Please revise causal language (“...indicating that short-term PM exposure (for even less than a day) could trigger pneumonia.”)

15. Discussion: Lines 293-294, Lines 299-300, Line 305,: Please revise causal language.

16. Discussion: Line 307: Please revise the text to avoid over-interpretation of the conclusion here.

17. Discussion: Line 362-363: Please revise the text to avoid causal language (...”assist in understanding how PM causes lung-inflammation…”)

18. Figure 1: Please revise Figure 1 such that it is possible to see the locations of the cities more clearly. Please clarify the meaning of the variation in dot size.

19. Figure 2: Please provide units for the axes.

20. Table 3 and Table 4, and S2, S3, and S4 tables: Please indicate if these analyses are adjusted for any factors, and if so, please specify which factors are being adjusted for, and provide unadjusted values.

21. Thank you for including the completed STROBE checklist as Supporting Information. When completing the checklist, please use section and paragraph numbers, rather than page numbers. Please add the following statement, or similar, to the Methods: "This study is reported as per the Strengthening the Reporting of Observational Studies in Epidemiology (STROBE) guideline (S1 Checklist)."

Comments from the reviewers:

Reviewer #1: I confine my remarks to statsitical aspects of this paper. These were very well done indeed and I have no problem recommending publication.

Peter Flom

Reviewer #2: The authors have conducted an interesting analysis on a health outcome that has not been well studied, which could aid in both the overall interpretation of PM effects on respiratory infections generally, but also specifically the potential health implications of PM exposures within China. Generally, the analysis is well conducted, but additional clarification is needed in many instances, and the authors should consider revising the approach used to examine effect modification. The following comments identify those areas and issues for the authors to consider in revising the manuscript:

- Throughout the manuscript there is too much reliance on statistical significance in presenting results from the study as well as in the discussion of studies in the introduction and discussion. The authors need to remove this and focus on patterns of associations. 

Abstract:

Line 28: I would specify that these are hospital admissions because I could see some getting confused as to whether these are hospital admissions or emergency department visits.

Line 32: should note here what effect modifiers will be examined, otherwise you don't know which ones until in the results section and seeing Table 5.

Line 41-42: This does not seem like something that should be in the abstract.

Introduction:

Line 69: This does not seem like something that should be in the abstract.

Line 72: This doesn't make sense to put here because this paper is focusing on short-term exposure and the estimates here are based on health impact functions for long-term PM2.5 exposures. It's not really making the correct point.

Line 74: should be association, not correlation.

Line 75: From this point on, there needs to be more context put around these studies versus just listing them out and what they found. Something like, some studies have started to report some evidence of an association between air pollution exposure (or PM2.5 specifically) and respiratory infections, including pneumonia. After that it will help in discussing these studies and their results. Also, it would be better to not just list studies, but find some commonality amongst them. 

Line 87: Throughout it says PM levels, but it should actually be PM concentrations. Need to fix this throughout the paper.

Line 88 - 92: This sentence is unclear and needs clarification.

Methods:

Line 98 - 101: This doesn't seem like the correct place for this statement. Also, should make it clearer that if cities did not collect the correct health data they were excluded from the analysis in these sentences.

Line 102 - 104: Should say the years of analysis differed by city. Also, it's not clear what is meant by 3-year data records, health data?

Figure 1: this map is confusing. Unable to clearly distinguish where the cities are located because so many dots. May be clearer to remove all names of provinces and only have dots for the cities. If you still want to detail the most populous cities it can be done in a supplemental table or have the size of the dots change depending on population of the city. Do the size of the dots mean something?

Study Population:

need to specify the age range examined because in the discussion it says adults, but not defined in the methods section.

Line 121: Should be legal instead of legally. 

Statistical analysis:

Line 162: what sensitivity analyses? if discussed later should note that.

Line 168 - 173: This discussion on the concentration-response relationship is unclear and could use a revision. It should be concentration versus exposure response. The last few lines of the paragraph needs additional explanation, it's unclear how 5 coefficients were examined and what the variance-covariance matrix represents. Additional explanation is needed to clarify how statements about the shape of the C-R curve where made.

Line 175: Would be better to say something like: We examined effect modification of the relationship between short-term PM exposure and pneumonia hospital admissions in analyses...

Line 178: The part about annual average PM levels is unclear and needs additional explanation. Actually, the whole discussion on how effect modification was examined should be reconsidered. The approach used in the paper is confusing. It would be easier to interpret results if the authors examined effect modification based on the distribution of the factor examined across cities. For example, like the NMMAPS or APHENA studies where they examined how the risk of mortality changed when moving from the 25th to 75th percentile of the factor examined.

Sensitivity Analyses:

Line 185 - 186: Clarify, does this mean the authors compared associations between cities with 3 years of data versus 4 years of data?

Line 187: Prior to the sentence starting "Fourth" : I would add a sentence prior to these last two analyses to note, that model specification was assessed using these two analyses.

Line 188: The sentence starting "Fifth", would be beneficial to start the sentence by saying: to examine whether we appropriately specified the regression model used in this analysis, we...

Results:

Line 216: should clarify whether single day lags or the average of 0-2 day.

Line 219: at the lag days examined instead of at different lag days.

Line 223: Separate paragraph should start here to talk about effect modification results.

Line 224: better to present the risk estimates and confidence intervals than to focus on P values.

Line 228 - 230: I don't recall this being mentioned earlier. How were these delineations defined? Need additional explanation.

Table 3: this table could be simplified by having the CIs directly next to the percent change. Don't need the P value column.

Table 4: it's really % increase, not % change, replace PC with % Increase

Line 240: based off the figures I wouldn't call the curves linear across the full range of concentrations examined. There is actually some evidence of supralinearity where the curve plateaus at some higher concentration. Granted those concentrations are above 100, but it tends to be consistent with some of the evidence from the Integrated Exposure Response Function. See Burnett et al. (2014) - EHP. This statement "in line with reports from other studies" is not entirely correct because it makes it sound like the other studies also examined pneumonia when there are mortality studies being cited. Could say instead that "this evidence of linearity is consistent with studies examined short-term PM exposure and other outcomes."

Line 245: should be concentration-response

Line 250: From this point forward, It's not clear how this analysis was done. I'm surprised the authors did not examine the distribution across all cities of these variables and then examine how the risk estimate may change when going from the 25th percentile to the 75th percentile of the factor being examined. Additionally, there is no discussion of the influence of annual PM concentrations on the results even though it is in the table. 

Line 268 - 269: Again, too much reliance on statistical significance, instead focus on whether the associations remained positive. Please revise to reflect this.

Line 269 - 271: I'm not following this discussion on C-R here. It's not clear. Also, I would not use the word threshold, you've really identified an inflection point where the risk changes due to non-linearities in the C-R curve.

Line 272 - 274: Similar comment as to above, group the model specification analyses together and note whether the results were consistent with the base model.

Line 277: clarify minor effects. Slightly attenuated, remained relatively unchanged?

Discussion:

Line 283: need to specify age range examined in the methods section.

Line 284: "acute effects", this is not correct, it's not the acute effects, but to investigate whether short-term PM exposures contribute to increases pneumonia hospital admissions.

Line 287: PM2.5 and PM10?

Line 288: short-time course? from studies examining short-term PM exposures.

Line 289: "for even less than a day", not sure how this is being deduced. We don't actually know this since the exposures being assigned are 24-h avg exposures and not hourly exposures.

Line 291: the evidence examining sub-daily exposures is very limited and this study cannot inform that since the authors are focusing on 24-h avg exposures. An association at lag 0 only indicates an immediate effect, it cannot inform that the effect is due to exposures of a shorter duration because the exposure metric used was 240h average, not individual hour values. Please revise.

Line 291 - 292: bringing in sub-daily cardiovascular results does not make sense, and again this study cannot inform sub-daily exposures because it is focusing on 24- avg exposures.

Line 292 - 294: a 0-2 lag is still considered an immediate effect. Delayed would be if you say associations at lag 2-5 days. Remove delayed and keep cumulative effect.

Line 296: Please see earlier comment on this characterization of the C-R curve.

Line 297 - 299: this is not the correct interpretation. if a health benefit per ton analysis was occurring sure, this statement could be made, but the analysis is focusing on risk estimates so the proper intepretation is that risk is higher and concentrations lower than some value and they plateau above a value. 

Line 299 - 300: This sentence is unclear and I don't think it's needed.

Line 302 - 304: Jumping to harvesting seems like a stretch. It could just be that PM2.5 has a greater health risk than PM10 and that PM10 is representing exposure to more particles in the coarse range (PM10-2.5) than fine particles. I don't think the harvesting statement makes sense based on this study. If the authors examined a longer duration of lags to support this statement then it would be a different story.

Line 305: But a threshold analysis was not conducted, the analysis only focused on assessing linearity. I don't think this statement is needed. If anything the authors could state the point at which confidence intervals expand and there is less certainty in the shape of the C-R at both lower and higher PM concentrations. 

Line 307: I'd remove statements about health benefits because this paper is not focusing on that and it is a little confusing. 

Line 309 - 311: Seems odd to cite a respiratory mortality study instead of a pneumonia HA one here. I would revisit other penumonia studies to confirm the findings of this study instead of mortality. 

Line 312-313: I think this is a better statement to make about differences between PM10 and PM2.5 versus the statement above about surface area. The makeup of the particles themselves is much different. 

Line 314: I would delete this sentence, especially since there is no citation.

Line 321-322: This paragraph is on the right tract, but it needs more at the end, the PM-metals part really doesn't fit in with the rest of the paragraph.

Line 324-325: I couldn't access the supplement. Did they stay positive? Need to better convey the results versus relying on statistical significance.

Line 325: From this point on, Too much reliance on statistical significance in this whole paragraph.

Line 330: Need to specify PM2.5.

Line 352-355: this needs additional explanation. Unclear what the authors are trying to convey.

Reviewer #3: This study investigates short term effects of air pollution on risk of pneumonia hospital admissions. This is a well conducted study and well written. Some issues need to be clarified, as outlined below.

Line 121: Can the authors remind the reader of the coverage rate of UEBMI?

Can authors elaborate further on using hospital admissions and not including emergency room visits. Hospital admissions will include the more severe cases only.

Statistical analysis section: Can authors provide a bit more details on how the degrees of freedom were chosen for the variables that were used in natural cubic splines? I suppose they looked at model fit, but this should be further explained.

Line 228: Authors refer to the online supplement when further stratifying regions, but can they provide a description of those findings in the text?

Can authors add the variables that were used for adjustment under each table? This is mentioned in the Statistical analysis section, but should also be reported under each table. For example, is Table 5 adjusted for temperature and humidity, in addition to looking at meta-regression models by city-level annual average temperature and humidity?

Did authors consider looking at multi-pollutant models (i.e. adjusting for more than two pollutants in their models)?

Line 276-277: Where are those results reported? Also, why using a cut-off of 20% for health insurance coverage?

Line 298: Can authors provide more discussion on why risks would be higher per unit increase at low levels? This is a very important topic going forward in air pollution epidemiology.

Line 316: Possible mechanistic pathways are outlined, but pneumonia is an infection and it seems there are missing discussion points on how air pollution could be increasing the risk of getting this type of infection. Further discussion is needed in this paragraph.

Can authors describe why they did not adjust for influenza epidemics in their models? This is a common approach when looking at short term air pollution-respiratory disease associations. 

Concentration-response curves: Can authors add the 24-hour standards for PM10 and PM2.5 as vertical dashed lines in the graphs? This would give a sense of what standards are currently and how effects are being observed in regards to those standards. 

Reviewer #4: The manuscript assesses the relationship between short-term exposure to particulate matter (PM) and hospital admissions for pneumonia in China for a 4-years period (2014-2017). The key finding is that PM is associated with increased pneumonia admissions, especially in the elderly population. Moreover, effects are stronger in cities with higher temperatures and relative humidity. 

The study bases on a huge dataset with data from 184 cities across China, representing differing climates across the country. Further, the question and the results are of interest to general practitioners and pulmonary specialists as well as public health officials; the topic is quite timely as air pollution is still affecting populations worldwide, even at places with low air pollution levels. The manuscript is well written and the statistical approach is appropriate. However, I have some concerns:

Major comments:

* My biggest concern is about the selection of pneumonia as the only outcome. For example, it would be interesting to look also at respiratory infections or influenza. Further, individuals with certain chronic medical conditions, such as COPD - which includes emphysema and chronic bronchitis - and asthma are especially at risk for pneumonia. Therefore, I strongly suggest including further outcomes.

* Statistical analysis, lines 152-161: As the association between PM and pneumonia might be confounded by influenza, I suggest adjusting for influenza in a sensitivity analysis.

* Statistical analysis, lines 163-166: Why did the authors choose lags of 0, 1 and 2 days? Previous studies have seen more delayed effects of PM, especially with respiratory diseases.

* Discussion, lines 324-335: The authors could also assess effect modification by other pollutants (NO2 or CO) using categories for pollutant levels.

* Discussion, lines 337-343: The authors need to provide a more detailed discussion and potential mechanisms for their finding of stronger effects in cities with higher temperature and relative humidity.

Minor comments:

1. Abstract, line 31: The authors should drop "overdispersed" here as the use of a quasi-Poisson model already implies consideration of overdispersion. Further, the authors should add here that they adjusted for confounding.

2. Study sites, lines 96-104: The authors should give more details on the choice/inclusion of cities. Why, for example, is Beijing not included, although it is the capital of China? What about Shanghai? What was the original number of cities? Further, authors should provide more details on what they mean with "Cities with no information on ICD code".

3. Study population: As the study uses data from the UEBMI system, results are not transferable to the rural population. This should be acknowledged in the limitations.

4. Lines 124-125: Please provide an ICD code here.

5. Environmental data: The authors state that they obtained hourly air pollution concentrations. How did they calculate 24-hour averages? Was the exclusion of "days with missing monitoring measurements" done for cities with only one monitoring site?

6. Lines 178-179: Please be more specific here and replace "weather conditions" with "air temperature and relative humidity".

7. Results: The authors need to give more details on the cities included. For example, what is the population of the cities? What is the UEBMI coverage per city? What is the distribution of daily hospital admissions, air pollutants and meteorological variables.

8. Figure 2: Please provide units for the axes. 

9. Table 5: Please mention the increment for the effects estimates.

10. Line 272: Why did the authors choose 100 μg/m³ as cut-off? The decline in the exposure-response function starts earlier.

[LINK]

---

## [Decision Letter · Decision Letter 1]

19 Nov 2019

Dear Dr. Hu,

Thank you very much for re-submitting your manuscript "Ambient Particulate Matter Pollution and Adult Hospital Admissions for Pneumonia in Urban China: A national time-series analysis" (PMEDICINE-D-19-02197R1) for review by PLOS Medicine.

I have discussed the paper with my colleagues and the academic editor and it was also seen again by 2 reviewers. I am pleased to say that provided the remaining editorial and production issues are dealt with we are planning to accept the paper for publication in the journal.

[LINK]

We look forward to receiving the revised manuscript by Nov 26 2019 11:59PM. 

Sincerely,

Caitlin Moyer, Ph.D.

Associate Editor 

PLOS Medicine

plosmedicine.org

Requests from Editors:

1.Prospective Analysis Plan: Thank you for including your pre-specified statistical analysis plan (S1 Appendix). If your statistical analysis plan was developed prospectively, please include text in the document to indicate the date the plan was developed. 

2. Thank you for your response to Reviewer 2, point 10. The sentence "Cities with only 1-year hospital admission records were excluded owing to the feasibility of model fit" is not clear. Please clarify what is meant by "feasibility of model fit".

3. Thank you for your response to Reviewer 2, point 16. However, the text "Variance-covariance matrix is a matrix whose element in the i, j position is the covariance between the i-th and j-th elements of a random vector. The regression coefficients derived from the main model with a B-spline function for PM" is missing from the manuscript. Please update the manuscript text to match your response.

4. Thank you for your response to Reviewer 2, point 18. However, please update the text of the methods as described in your response.

5. Thank you for your response to reviewer 2, point 21. However, please update the text as you describe in your response.

6. Thank you for your response to reviewer 2, point 27. However, please present both 95% CIs and p values for your results, according to PLOS Medicine policy. Please also do this throughout the manuscript.

7. Thank you for your response to Reviewer 2, point 44. However, the statement "the curves plateau at high concentrations, indicating that hte risk of pneumonia admission is lower at high concentrations." is confusing. Does this sentence mean that the increase in risk of pneumonia admission is greater at lower compared to high concentrations? If so, please clarify.

8.Thank you for your response to reviewer 2, point 47. However, deleting the term "harvesting" does not adequately address the reviewers point that there may be different health risks associated with different PM sizes, or other explanations.

9. Title: Please revise the title to: “Ambient Particulate Matter Pollution and Adult Hospital Admissions for Pneumonia in Urban China: A national time-series analysis for 2014 through 2017.” or similar, as we would prefer to have the dates of your study included in the title to provide time relevance.

10. Abstract: Background: Line 29-30: Please revise to, “We aimed to examine the association between PM levels and hospital admissions for pneumonia in Chinese adults.” or similar, to reflect that your study did not measure individuals’ exposures to PM.

11. Abstract: Conclusions: Line 51-52: Please revise to, “Our findings suggest that there are significant short-term associations between ambient PM levels and increased hospital admissions for pneumonia in Chinese adults.” or similar to reflect that you were not evaluating individual exposures.

12. Abstract: Conclusions: Please revise to, “These findings support the rationale that further limiting PM concentrations in China may be an effective strategy to reduce pneumonia-related hospital admissions.” or similar.

13. Author Summary: “Why was this study done?”: Here and throughout the text of the manuscript, please refer to low or middle income countries rather than "developing countries". Please refer to high income countries rather than "developed" countries. 

14. Author Summary: “What did the authors do and find?”: Please combine the first two bullet points into a single point such as: “We conducted a nationwide time-series analysis using data on more than 4.2 million hospital admissions for pneumonia in 184 cities in China between 2014 and 2017 to estimate city-specific, and national and regional average associations between ambient PM pollution and pneumonia hospitalizations."

15. Author Summary: “What did the authors do and find?”: Please replace “exposures to” with “increases in” for the third bullet point.

16. Author Summary: “What do these findings mean?”: Please revise the first bullet point to “...short term associations of PM levels with hospital admissions…”

17. Author Summary: “What do these findings mean?”: Please revise the second bullet point to: “Our findings support the rationale for further limiting PM concentrations in low-middle income countries. -or similar.

18. Introduction: Line 115-117: Here, and throughout the text of the manuscript, please replace “developed countries” with “high income countries”, and “developing countries” with “low or middle income countries”. 

19. Introduction: Line 123-124: Please revise to, “In this study, we examined the short term associations between concentrations of ambient PM pollution and daily hospital admissions for pneumonia in adults in China between 2014 and 2017.” to make the goal of the study clear.

20. Methods: Line 133-134: “Cities with no information on disease diagnosis recorded in the database were also excluded” Following the comments of Reviewer 4, this may be a good place to clarify some cities excluded, including Beijing and Shanghai.

21. Methods: Lines 222-223: Please revise to: “We examined effect modification of the relationship between short-term ambient PM concentrations and pneumonia hospital admissions in analyses by sex, age, and region…”

22. Methods: Lines 246-249: For statistical testing, please specify the significance level used (e.g., P<0.05, two-sided) and the statistical test used to derive a p value.

23. Results: Line 265 and Table 2 title: Please change “exposure variables” to “environmental” or “air quality” variables.

24. Results: Line 265-269: Please provide p values and accompanying r values for the reported correlations between environmental/air quality variables.

25. Results: Line 276-282: Please provide 95% CI and p values for the percent increases of hospital admissions for pneumonia with increases in concentrations of air quality measures.

26. Results: Lines 288-291: For the comparisons between males and females, please provide the specific p values, and also provide values for each comparison (PM 2.5 and PM 10).

27. Results: Lines 293-297: Please provide p values associated with differences between the regions for each PM 2.5 and PM 10.

28. Results: Lines 291-294: Please provide p values associated with comparisons between age groups.

29. Results: 346-352: Please provide p values associated with the 95% CIs reported for these analyses.

30. Results: 359-360, and throughout the manuscript: Please provide the specific p value in place of [NS or P>.05].

31. Discussion: Line 374: Please revise this to “...higher pneumonia-related hospital admissions…”

32. Discussion: Line 376: Please revise this to: “...investigate whether short-term changes in ambient PM concentrations are related to increases pneumonia hospital admissions…”

33. Discussion: Line 379: Please revise this to: “We observed increased risk of hospital admissions for pneumonia in association with both PM2.5 and PM10…”

34. Discussion: Line 411: Please revise this to: “...findings on the short-term association between ambient PM concentrations and pneumonia hospitalizations…”

35. Discussion: Line 442: Please revise this to: “This study provided detailed estimates of the risk of PM-associated hospital admissions for pneumonia through the use of data…”

36. Discussion: Line 466-468: Please revise this to: In summary, we found that short-term elevations in PM were associated with increased pneumonia- related hospital admissions in Chinese adults. Our findings support the rationale to further limit PM concentrations in China.

37. Table 1, 2, 3, 4 and 5: Please define abbreviations for “PM2.5” and “PM10” in the legends.

38. Table 2: Please provide p values associated with all statistical comparisons, include actual values for p<0.01, unless p<0.001.

39. Table 3: Please provide both the 95% CIs and p values for these results.

40. Table 3: Please also provide the results for the unadjusted analyses.

41. Table 4: Please also provide the results for the unadjusted analyses.

42. Table 4: Please provide p values to accompany the 95% CI for all comparisons shown.

43. Table 5: Please replace “P” with “P value”

44. Figure 2: To facilitate comparisons, please display the two graphs on the same y-axis scale, or please note in the legend that the y-axes of the two graphs are scaled differently.

45. S2 Table, S3 Table, S4 Table, S5 Table: Please also provide results for unadjusted analyses. Please provide p values to accompany the 95% CIs for these analyses.

Comments from Reviewers:

Reviewer #3: Authors have made tremendous efforts to address all comments. I have no issues in recommending publication.

Reviewer #4: The manuscript assesses the relationship between short-term exposure to particulate matter (PM) and hospital admissions for pneumonia in China for a 4-years period (2014-2017). The key finding is that PM is associated with increased pneumonia admissions, especially in the elderly population. Moreover, effects are stronger in cities with higher temperatures and relative humidity. 

The study bases on a huge dataset with data from 184 cities across China, representing differing climates across the country. Further, the question and the results are of interest to general practitioners and pulmonary specialists as well as public health officials; the topic is quite timely as air pollution is still affecting populations worldwide, even at places with low air pollution levels. The manuscript is well written and the statistical approach is appropriate. 

I appreciate the effort the authors have spent on addressing the concerns raised in the initial review. The modifications clearly have improved the quality and clarity of the manuscript.

Minor comments:

1. Study sites, lines 96-104: Although the authors now give more details on the inclusion criteria, it is still unclear why, for example, Beijing is not included, although it is the capital of China? Same is true for Shanghai. 

2. Lines 390-392: The authors speculate that "The leveling-off at high concentrations might be speculated by that people vulnerable to PM exposure may have developed symptoms and sought treatment before PM concentrations reached high concentrations". Other reasons could be that people avoid spending time outdoors or start wearing facemasks.

[LINK]

---

## [Editor Report · Decision Letter 2]

4 Dec 2019

Dear Prof. Hu, 

On behalf of my colleagues and the academic editor, Dr. Aziz Sheikh, I am delighted to inform you that your manuscript entitled "Ambient Particulate Matter Pollution and Adult Hospital Admissions for Pneumonia in Urban China: A national time-series analysis for 2014 through 2017" (PMEDICINE-D-19-02197R2) has been accepted for publication in PLOS Medicine. 

PRODUCTION PROCESS

PRESS

PROFILE INFORMATION

Thank you again for submitting the manuscript to PLOS Medicine. We look forward to publishing it. 

Best wishes, 

Caitlin Moyer, Ph.D.

Associate Editor 

PLOS Medicine

plosmedicine.org